# The Relationship Between Bacterial Flora in Saliva and Esophageal Mucus and Endoscopic Severity in Patients with Eosinophilic Esophagitis

**DOI:** 10.3390/ijms26073026

**Published:** 2025-03-26

**Authors:** Momoyo Sasahira, Hiroshi Matsumoto, Tei Tei Go, Shogen Yo, Shuzo Monden, Takahirao Ninomiya, Motoyasu Oosawa, Osamu Handa, Eiji Umegaki, Ryo Inoue, Akiko Shiotani

**Affiliations:** 1Department of Gastroenterology, Kawasaki Medical School, Okayama 701-0192, Japan; momomo0318@gmail.com (M.S.); gutingting0529@med.kawasaki-m.ac.jp (T.T.G.); yoshogen@med.kawasaki-m.ac.jp (S.Y.); mondymondy@gmail.com (S.M.); takehiro.nino0614@gmail.com (T.N.); o.m.1976-1017@med.kawasaki-m.ac.jp (M.O.); handao@med.kawasaki-m.ac.jp (O.H.); eumegaki@med.kawasaki-m.ac.jp (E.U.); shiotani@med.kawasaki-m.ac.jp (A.S.); 2Faculty of Agriculture, Setsunan University, Hirakata-City 573-0101, Japan; ryo.inoue@setsunan.ac.jp

**Keywords:** eosinophilic esophagitis, saliva microbiota, esophageal microbiota, mucosa-associated microbiota, bacterial flora

## Abstract

Eosinophilic esophagitis (EoE) is a chronic inflammatory disease characterized by esophageal dysfunction and eosinophilic inflammation of the esophageal mucosa. In this study, we investigated the bacterial flora in saliva and esophageal mucus in patients with EoE and examined the relationship between EoE disease activity and mucosal cytokine expression, involving patients with active and inactive EoE (A-EoE and I-EoE, respectively). A-EoE was defined as a peak eosinophil count > 15/high-power field, according to the 2025 consensus guidelines. Saliva samples were collected from patients before the endoscopic examination. Brushing samples were collected from the distal esophagus of patients with EoE during endoscopic procedures. The degree of EoE inflammation was assessed using the EoE endoscopic reference score (EREFS). The samples were profiled using the Illumina MiSeq platform. The V3–V4 regions of the 16S rRNA gene (460 bp) were amplified using tailed PCR. Fifty-nine patients were enrolled, including eight with I-EoE, seventeen with A-EoE, and twenty-eight non-EoE controls. Major bacterial genera such as *Streptococcus*, *Prevotella*, *Veillonella*, and *Haemophilus* were detected in both the oral cavity and esophagus. Compared with the control group, the active EoE group had significantly more *Prevotella* spp. in the saliva and esophageal mucosa. Conversely, significantly fewer *Neisseria* spp. were found in the saliva and *Streptococcus* spp. in the esophageal mucosa of patients with active EoE. The EREFS of EoE and *Streptococcus* were inversely correlated. This study elucidated the characteristics of bacterial flora in the saliva and esophageal mucosa of patients with EoE.

## 1. Introduction

Eosinophilic esophagitis (EoE) is a chronic inflammatory disease characterized by esophageal dysfunction and eosinophilic inflammation of the esophageal mucosa [1,2,3,4,5]. The incidence and prevalence of EoE have increased dramatically over the past 20 years. When EoE was first reported in 1977 [6], the incidence was only approximately 0.4 cases per 100,000 people per year [7]. Now, the incidence rate has increased to approximately 10 cases per 100,000 people per year, with the estimated prevalence rate ranging between 50 and 100 cases per 100,000 people [8]. The increased incidence and prevalence of EoE have been observed worldwide, with prevalence tending to be higher in Western countries than in Asian countries. In particular, the prevalence of EoE in North America and Europe is significantly higher than that in Asia. However, the incidence of EoE is also increasing in Asian countries, requiring further investigation into the relevant factors. This increase is more pronounced in high-income countries, with a higher incidence and prevalence in males, especially in boys.

The etiology and pathogenesis of EoE have not yet been fully understood, but several hypotheses have been proposed based on current findings. EoE often runs in families, suggesting a genetic predisposition [9]. Specific genes may increase the risk of EoE [10,11]; however, the specific genes and their mechanisms remain unclear. Furthermore, EoE is associated with abnormal immune responses to foods and environmental allergens [12]. The function of the esophageal epithelial barrier is impaired in patients with EoE [13,14]. Dysfunction of the esophageal epithelial barrier could facilitate the penetration of allergens and microorganisms into the esophageal mucosa, causing inflammation. In particular, the overproduction of Th2 cytokines (such as IL-4, IL-5, and IL-13) may play an important role in eosinophil infiltration and esophageal inflammation [15]. In EoE, the amount of bacteria and specific Toll-like receptors are overexpressed compared with controls. Furthermore, since they normalize with a diet-elimination method, microbial involvement may be strongly related to disease pathogenesis [16].

Oral bacterial flora is a highly complex living environment in the body that regulates the host’s immune system and may contribute not only to oral diseases but also systemic diseases [17]. The relationship between salivary microbiome and EoE has recently attracted attention [18,19]. The oral cavity is an important entry point for triggering allergens, which may subsequently interact and mix with saliva components (including the saliva microbiome) to trigger an immune response that results in eosinophil-dominant esophageal inflammation. Several studies have reported altered salivary microbiome composition in patients with EoE, which may be associated with inflammation. Furthermore, although the mechanisms are yet to be proven, bacteria in saliva may migrate to the esophagus and alter the esophageal microbiome.

A diverse bacterial flora also develops in the esophageal mucosa. Recent research has focused on changes in the bacterial flora caused by esophageal diseases such as reflux esophagitis and esophageal cancer (adenocarcinoma and squamous cell carcinoma) [20]. Changes in the esophageal microbiome can also influence EoE development and pathogenesis [9,21]. Changes in the composition and diversity of esophageal bacterial flora have been reported in patients with EoE, with increases or decreases in specific bacteria, such as *Haemophilus* and *Streptococcus* [9,22]. These changes in the bacterial flora may alter the immune response of the esophageal mucosa, inducing the production of Th2 cytokines and chemokines [3]. Furthermore, esophageal microbiome alterations may also affect the function of the esophageal epithelial barrier. However, many aspects remain unknown, such as the relationship between changes in esophageal mucosal flora and the severity of EoE as well as the relationship between cytokines.

In this study, we investigated the bacterial flora in saliva and esophageal mucus in patients with EoE and examined the relationship between EoE disease activity and mucosal cytokine expression.

## 2. Results

### 2.1. Patient Characteristics

The demographic and clinical characteristics of the entire study population are presented in Table 1. Of the 59 patients enrolled, 25 were EoE patients (8 patients of I-EoE and 17 patients of A-EoE), and 28 were non-EoE controls. The groups were comparable for age (EoE patients’ mean age: 51, interquartile range (IQR): 40–57 years’ vs. non-EoE patients’ mean age: 49, IQR: 29–62 years; *p* = 0.13). Dysphagia (41%) and abdominal pain (26%) were two of the most common indications for esophago-gastroduodenoscopy (EGD). A significantly higher proportion of patients with active EoE presented with dysphagia compared with non-EoE controls (76% vs. 14%; *p* < 0.01). Food or drug allergies (35%) and allergic rhinitis (33%) were the most common allergic comorbidities. Moreover, 11 (44%) EoE patients were taking proton pump inhibitors (PPIs), 4 (16%) EoE patients were taking topical steroids, and 13 (52%) patients were taking antiallergic medicines such as fexofenadine. A significantly higher proportion of patients with I-EoE took PPIs compared with A-EoE (88% vs. 24%; *p* = 0.01) (Appendix A). ERFES scores with A-EoE were significantly higher compared with I-EoE (mean: 2, IQR: 4–4 vs. mean: 2, IQR 0–2). Further, six (21%) non-EoE patients were taking PPIs for gastroesophageal reflex disease (GERD) or abdominal pain.

### 2.2. Comparative Analyses of the Alpha Diversity Indices Among the Three Groups Using Saliva and Esophageal Mucus

There were no differences in Shannon, Chao1, or observed feature indices for the alpha diversity of saliva (Figure 1A–C). On the other hand, the Shannon index was significantly lower in the esophageal mucosa in the non-EoE patients (Figure 1D). However, no significant difference was observed in the other index, Chao1, or observed features in the esophageal mucosa (Figure 1E,F).

### 2.3. Comparative Analyses of Saliva Taxonomic Composition

Figure 2 shows a bar plot of the top 10 genera in the bacterial flora of saliva (Figure 2A) and the esophagus (Figure 2B). Major genera such as *Streptococcus*, *Prevotella*, *Veillonella*, and *Haemophilus* were found in both the oral cavity and the esophagus. *Streptococcus* was found more frequently in the esophagus than in the oral cavity, which is consistent with previous reports. In saliva, in a comparison of the two groups, the EoE group had significantly more *Prevotella* and *Anaeroglobus* than the control group, while *Neisseria* and *Haemophilus* were significantly less common (Appendix A). In the esophageal mucosa, in a comparison of the two groups (control group and EoE group), the EoE group had significantly more *Prevotella* than the control group and, conversely, significantly less *Streptococcus*.

In saliva, a comparison among the three groups revealed a significant difference in *Prevotella* spp. and *Neisseria* spp. (Figure 3A,B). The number of *Prevotella* spp. was significantly lower in the control and I-EoE groups than in the A-EoE group (*p* < 0.05, d < 0.5; control group 0.18 ± 0.07 [0.15–0.21], I-EoE 0.25 ± 0.03 [0.23–0.28], A-EoE 0.24 ± 0.07 [0.20–0.28]). *Neisseria* spp. counts were significantly lower in the A-EoE group than in the control and I-EoE groups (*p* < 0.05, d < 0.5; control group 0.07 ± 0.05 [0.05–0.09], I-EoE 0.03 ± 0.02 [0.01–0.05], A-EoE 0.04 ± 0.03 [0.02–0.05]).

Figure 3C–E show a comparison of salivary samples between the two groups. In the comparison between the control and I-EoE groups, *Prevotella* was significantly more abundant in the I-EoE group than in the control group (C). In the comparison between the control and A-EoE groups, *Prevotella* was significantly more common in the I-EoE group than in the control group, whereas *Neiseria* and *Hemophilia* were significantly less common (D). In the comparison between the I-EoE and A-EoE groups, the abundance of *Bacteroides* was significantly more common in the I-EoE group than in the A-EoE group (E).

### 2.4. Comparative Taxonomic Analysis of the Esophageal Mucosa

In the esophagus mucus, a comparison among the three groups revealed significant differences in *Streptococcus* spp. and *Prevotella* spp. (Figure 2B and Figure 4A,B). *Streptococcus* spp., the most common genus in the esophagus, were significantly less common in the active EoE group compared to the inactive EoE and non-EoE groups (*p* < 0.05, d < 0.5; control 0.41 ± 0.18 [0.32–0.49], I-EoE 0.33 ± 0.11 [0.23–0.42], A-EoE 0.21 ± 0.10 [0.16–0.26]). *Prevotella* spp. were significantly more common in the active EoE group compared to the inactive EoE group (*p* < 0.05, d < 0.5; control 0.18 ± 0.07 [0.06–0.15], I-EoE 0.26 ± 0.03 [0.14–0.21], A-EoE 0.24 ± 0.07 [0.14–0.22]).

Figure 4C–E show the comparison of esophageal samples between the two groups. In the comparison between the control and I-EoE groups, *Prevotella* was significantly more abundant in the I-EoE group than in the control group (C). In the comparison between the control and A-EoE groups, *Prevotella* was significantly more common in the I-EoE group than in the control group, whereas *Streptococcus* was significantly less common (D). In the comparison between the I-EoE and A-EoE groups, *Streptococcus* was significantly more common in the I-EoE group than in the A-EoE group (E).

### 2.5. Correlation Between Salivary and Esophageal Flora

As a result of examining the correlation between the bacterial flora of saliva and esophageal mucosa, no correlation was found for *Streptococcus* (Figure 5A), but a correlation was found for all of the other top bacterial species: *Prevotella* (Figure 5B), *Haemophilus*, *Veillonella*, *Gemella*, *Alloprevotella*, *Fusobacterium*, *Pauijensenia*, *Porphyromonas*, *Neisseria*, and *Rothis*. In particular, *Prevotella* was strongly correlated.

We examined differences in bacterial abundance for each ERFFS item (box-and-whisker diagram, Figure 6C). The lower the ERFFS score, the greater the amount of *Streptococcus* present, and a significant difference was observed in particular for edema, rings, and furrows.

## 3. Discussion

This study investigated the microbiome of saliva and esophageal mucus in patients with EoE. Compared with the control group, significantly more *Prevotella* spp. were observed in the saliva and esophageal mucosa of patients with active EoE. Conversely, we found significantly fewer *Neisseria* in the saliva and *Streptococcus* in the esophageal mucosa of patients with active EoE. The endoscopic findings of EoE and the abundance of *Streptococcus* were inversely correlated, indicating a specific association with edema and longitudinal furrow.

*Prevotella* are commonly found in the gut and mouth, and their presence varies according to dietary habits and the living environment. In this study, *Prevotella* in the saliva and esophageal mucosa were significantly abundant in active EoE. The increased abundance of *Prevotella* spp. may be involved in the onset and progression of autoimmune diseases such as rheumatoid arthritis, systemic lupus erythematosus, and multiple sclerosis [17,23,24]. The abundance of *Prevotella* spp., specifically *Prevotella salivae*, in the saliva of patients with IBD was higher than that in healthy controls [25]. *Prevotella* activates the immune system and specifically promotes the secretion of inflammatory cytokines (e.g., IL-17 and IL-23) via Th17 cells. The *Prevotella*-Th17/IL-23 axis has a significant role in chronic inflammatory diseases [26]. Certain *Prevotella* strains, such as *P. histicola*, exhibit anti-inflammatory properties [27], whereas other strains, including *P. copri* and *P. nigrescens*, induce inflammation via IL-23-mediated activation of Th17 cells [28,29,30]. Although the Th17/IL-23 axis and eosinophilic inflammation (eosinophilic inflammation, EI) belong to different immune pathways, their interaction is attracting attention. IL-23 may be involved not only in the activation of Th17 cells but also in the activation of eosinophils and tissue infiltration, and in some diseases, where Th17 and Th17 and Th2 coexist (asthma, eosinophilic chronic rhinosinusitis (ECRS), atopic dermatitis, inflammatory bowel disease (IBD), etc.), inflammation tends to worsen, and steroid resistance may occur [31,32,33,34]. *Prevotella* produces lipopolysaccharide and other metabolites that may cause systemic inflammation and tissue damage [17,35]. In EoE, allergen-present naive T cells differentiate into Th17 cells, which secrete inflammatory cytokines and induce humoral and cellular factors. Therefore, an increase in the abundance of *Prevotella* spp. may contribute to the pathogenesis of EoE [36]. They may also play an important role in EoE, but further research is needed to elucidate their pathogenesis and the associated mechanisms. Further research will be required to examine more closely the association between these factors in eosinophilic gastrointestinal disorders.

The genus *Streptococcus* is an important commensal bacterial genus in the human oral and pharyngeal cavities and includes species that are pathogenic under certain conditions. Some *Streptococcus* spp. form biofilms and evade host defense mechanisms to proliferate infection. In both the present and previous studies, *Streptococcus* spp. were significantly decreased in the esophageal mucosa. The study by Zhang et al. found that some bacterial species in the esophageal microbiome, particularly *Streptococcus* spp., decreased in patients with EoE compared with healthy individuals [21]. Interestingly, some streptococci have been considered as potential probiotics. The *Lactobacillales*, to which *Streptococcus* belongs, may have antiallergic and anti-inflammatory effects. Supplementation with lactic acid bacteria attenuated pathogenicity and reduced esophageal eosinophilia in an EoE mouse model [37]. *Streptococcus salivarius* K12 is a probiotic that helps prevent upper respiratory tract infections. This bacterium produces antimicrobial peptides (bacteriocins) that may suppress harmful bacteria in the mouth and throat by competitive exclusion [38]. *S. salivarius* can also help prevent tooth decay and periodontal disease and demonstrates a high degree of oral adhesion [39]. It has been used in oral care products (lozenges and mouthwashes) and may help reduce the risk of tooth decay. Bacteria of the *Streptococcus* genus, and in particular *Streptococcus salivarius*, have attracted attention in recent years for their potential as probiotics. They have been reported to have anti-inflammatory effects against periodontal disease [40] and to be effective against periodic fever, aphthous stomatitis, pharyngitis, and cervical adenitis (PFAPA) syndrome [41]. In the present study, the number of *Streptococcus* spp. was significantly lower in the active EoE group and negatively correlated with the EREFS score, which may thus reflect the severity of EoE. *Streptococci* may be effective probiotics for EoE and should be further investigated.

The composition of the salivary microbiome is altered in patients with EoE. Nonpathogenic species, such as *Neisseria cinerea* and *Neisseria sicca*, which are known commensals of the oral cavity and upper respiratory tract, are part of the healthy oral microbiome. They help maintain oral and respiratory health by contributing to the production of antioxidants to mitigate oxidative stress and inhibit the growth of other pathogenic bacteria [42,43]. Furthermore, the present study found significantly fewer *Neisseria* spp. in the saliva of patients with A-EoE. A decrease in *Neisseria* spp. was also reported in the mouths of patients with recurrent aphthous stomatitis. This change in the *Neisseria* subpopulation was associated with the local inflammatory milieu, indicating the involvement of *Neisseria* spp. in the regulation of inflammation.

The relationship between oral and esophageal microbiomes has been investigated in the context of various diseases. Bacteria in saliva may migrate to the esophagus through the swallowing mechanism and subsequently induce changes in the esophageal microbiome. Changes in the esophageal microbiome may cause an immune response in the esophageal mucosa, which may promote inflammation in EoE [9]. In patients with EoE, inflammation may also occur in the mouth, which in turn may affect the composition of the salivary microbiome. EoE may also be a systemic immune disorder. In the present study, the *Prevotella* subpopulation was significantly increased in both the saliva and esophageal mucosa. Interestingly, the *Prevotella* subpopulation is also increased in the mouths and intestines of patients with inflammatory bowel disease (IBD), which can also develop when oral bacteria from human patients with IBD are transplanted into mice [26,44]. In other words, *Prevotella* in the mouth may influence the development of inflammation throughout the digestive tract and the whole body. However, the mechanism is not yet well understood. Further research is required to elucidate these details.

It has been posited that the salivary microbiota may serve as a noninvasive biomarker for eosinophilic esophagitis (EoE) [19]. A comparative analysis of the salivary microbiota composition between children with EoE and non-EoE has revealed alterations in the former group compared with the latter. Specifically, children diagnosed with EoE exhibited lower relative abundances of *Streptococcus*, *Lactobacillus*, *Leptotrichiaceae* (unclassified), and *Actinomyces* compared with children in the non-EoE control group. In contrast, the relative abundance of *Streptococcus* was significantly higher in the control group. Children with active EoE exhibited lower relative abundances of *Actinomyces*, *Lactobacillus*, *Leptotrichiaceae* (unclassified), and *Streptococcus* than non-EoE subjects. In the control group, the relative abundance of *Neisseriaceae* (unclassified) in non-EoE subjects was significantly higher. These findings also suggest that bacterial flora may discriminate between patients. In this study, the composition of the salivary microflora was similar between the EoE and non-EoE groups. Similar results were shown in this study for *Streptococcus* and *Neisseriaceae*. However, it is necessary to further investigate whether changes in the salivary microbiome can be used to help diagnose EoE, determine responsiveness to treatment, and predict prognosis.

*Streptococcus* in the esophageal mucosa was found to be correlated with EoE endoscopic scores, particularly in patients with edema, longitudinal ulcers, and furrows. This study is the first to detail the relationship between the EoE endoscopic scores and bacterial flora. *Streptococcus* is involved in the intestinal barrier and has been indicated as a potential probiotic; thus, the results of this study are confirmatory. However, in children with active EoE, the relative abundance of *Haemophilus* was significantly higher, and the relative bacterial abundance was significantly correlated with the abnormal esophageal mucosa and histopathological severity as indicated by the EoE histological score [19]. The study subjects’ young age may be the reason for the varying results. Many studies on the differences between the EoE and normal intestinal flora have mostly involved pediatric patients. The intestinal flora changes with age; thus, differences in intestinal flora between children and adults should also be assessed.

It has been posited that alterations in the esophageal microbiota of patients may exert an influence on specific metabolic pathways associated with EoE [45]. These alterations may contribute to the onset and progression of the disease by affecting sulfur compounds, 4-aminobutyric acid, and ornithine–arginine metabolism. An increase in EoE has been observed, and Gram-negative bacteria samples have been described as possibly affecting the inflammatory process [46]. The cell wall components of Gram-negative bacteria have been identified as stimulants of host immune receptors (e.g., Toll-like receptors) and inducers of inflammatory responses. It has been proposed that may play a role in the pathway mediated by Toll-like receptor (TLR) activation of inflammatory responses in EoE [21]. It has been proposed that alterations in the microbiota, achieved through the modulation of TLR-mediated immune responses, may exert an influence on the pathophysiology of EoE. Unfortunately, this study did not assess functional microbiota changes, including metagenomic or metabolomic changes. This necessitates further study.

This study has several limitations. First, the number of patients was small. The small sample size limits the generalizability of the results and increases the risk of type II errors. Second, the study was single-center, which reduces external validity. In the future, multicenter validation studies should be considered. Third, the study characterized bacterial composition but did not assess functional microbiota changes, such as metabolic activity or microbial interactions with the host immune system. Therefore, future studies should involve combined investigations into microbial communities and metabolomics for verification. The esophageal biopsies were taken only from the lower esophagus; however, different sampling sites may produce conflicting results. Sampling only the distal esophagus may miss regional microbial variations. Furthermore, several factors could influence microbiome composition, including dietary habits, antibiotic use, proton pump inhibitor (PPI) use, and genetic predisposition. The study did not control for these confounders, making it difficult to determine whether the observed microbiota differences were causative or incidental. In order to control for these confounding factors, it will be necessary to establish more rigorous selection criteria, conduct stratified analyses based on the use or non-use of PPIs used in treatment, and conduct longitudinal studies of the same participants’ microbiomes before and after treatment. By integrating these approaches, we must persist in our endeavors to comprehend the impact of confounding factors such as diet, antibiotic utilization, PPI use, and genetic predisposition on the outcomes of esophageal microbiome research in EoE, thereby facilitating the attainment of more reliable conclusions.

In this study, we clarified the characteristics of the bacterial flora in the saliva and esophageal mucosa of patients with EoE. However, further research is needed to determine whether these changes in bacterial flora are the cause or effect of EoE. Furthermore, their utility in diagnosis and treatment should also be evaluated.

## 4. Materials and Methods

### 4.1. Ethics

Ethics approval was obtained from the Kawasaki Medical School Ethics and Medical Research Committee (approval no. 5436-01). Written informed consent was obtained from all study participants before enrollment. All patients were treated at the Division of Gastroenterology of Kawasaki Medical School Hospital.

### 4.2. Patients and Sample Collection

We recruited patients with EoE or with symptoms of esophageal dysfunction who underwent esophagogastroduodenoscopy (EGD) at our center between April 2021 and May 2024. The exclusion criteria were inflammatory bowel disease, including non-EoE EGD; a history of esophageal surgery, varices, or esophageal cancer; or use of or exposure to antibiotics within the previous 4 weeks (Table 2).

Cases consisted of patients with active and inactive EoE (A-EoE and I-EoE). A-EoE was defined as a peak eosinophil count > 15/high-power field (hpf), as recommended by the 2025 consensus guidelines [47], whereas inactive disease was defined as a peak eosinophil count < 15/hpf. Biopsies of at least two samples were taken from the distal esophagus. The control group consisted of patients with non-EoE with symptoms suggestive of esophageal dysfunction but normal esophageal histology. We collected brush samples from the distal esophagus mucosa using an endoscopic microbiology brush (COOK, Bloomington, IN, USA) to prevent contamination. The brush method was selected as the specimen sampling method based on its reportedly higher bacterial DNA recovery rate than that by biopsy [48]. Saliva samples were collected from patients before the endoscopic examination.

The EoE endoscopic reference score (EREFS) is an indicator for evaluating the degree of inflammation in the esophagus [47,49,50]. The EREFS is based on five endoscopic findings, namely exudates, rings, edema, furrows, and strictures, and it is useful for evaluating EoE severity (Figure 7).

### 4.3. RNA Extraction

Esophageal tissue samples were collected from the distal esophagus simultaneously with the brush samples. The tissue biopsies were frozen at −80 °C until analysis. Total RNA was extracted from the esophagus using a NucleoSpin RNA kit and reverse-transcribed using the SuperScript™ IV First-Strand Synthesis System, according to the manufacturer’s instructions.

### 4.4. Reverse Transcription Quantitative Polymerase Chain Reaction

Reverse transcription quantitative polymerase chain reaction (RT-qPCR) was performed using PowerUp^TM^ SYBR^TM^ Green Master Mix (Thermo Fisher Scientific, Waltham, MA, USA). Human actin-beta expression was evaluated as an internal control. The primers used for the RT-qPCR experiments are listed in Table 1. The PCR conditions were as follows: initial denaturing at 95 °C for 2 min; 40 cycles at 95 °C for 15 s; 60 °C for 1 min; and melting curve analysis (95 °C for 15 s, 60 °C for 1 min, and 95 °C for 15 s). The primers used are listed in Table 2.

### 4.5. DNA Extraction, PCR Amplification, Library Preparation, and 16S rRNA Sequencing

The 16SrRNA gene amplicon library was prepared and sequenced using previously reported methods for DNA extraction, amplicon generation, and sequencing [51]. A library of amplicons encoding the 16S rRNA gene was prepared, and sequencing was conducted as previously described [52]. The samples were profiled using high-throughput amplicon sequencing with dual-index barcoding using the Illumina MiSeq platform (Illumina, San Diego, CA, USA). The V3–V4 regions of the 16S rDNA, which span 460 base pairs, were amplified using tailed PCR [53]. PCR amplicons were purified using SPRI select beads (Beckman Colter, Brea, CA, USA). The DNA concentration of the purified amplicons was measured using a Quantus Fluorometer and the QuantiFluor^®^ dsDNA System (Promega, Madison, WI, USA), and an approximately equal amount of their DNA was pooled. The pooled sample was sequenced using a MiSeq Reagent Kit V3 (600 cycles; Illumina, San Diego, CA, USA) on the MiSeq system according to the manufacturer’s instructions.

### 4.6. Bioinformatics Analysis

Sequence data processing, consisting of quality filtering, chimera checking, and amplicon sequence variant (ASV) definition, was accomplished using the QIIME 2. ver 2024.5 (https://qiime2.org/). Singletons were eliminated, and ASVs were taxonomically assigned using a scikit-learn naïve Bayes machine-learning classifier trained on the Greengenes 2. ASVs assigned to chloroplast and mitochondria were removed for this study.

### 4.7. Alpha Diversity

The observed features, Shannon entropy, and Chao1 diversity indices were calculated using the phyloseq package of R software (version 4.4.1). The data were visualized by principal coordinate analysis and statistically analyzed using permutational multivariate analysis of variance (PERMANOVA), which was performed using the vegan package (version 2.6-2).

### 4.8. Statistical Analysis

The differences in families among the lines were analyzed using the Statistical Analysis of Metagenomic Profiles software (version 2.1.3) [54]. The diversity and relative abundance of bacterial genera indices were compared among the three groups by the Kruskal–Wallis analysis and compared between the two groups by the Mann–Whitney U test. The category data were analyzed using the chi-square test. The correlation between saliva and esophageal bacterial flora and the correlation between bacterial flora and the ERFES score were evaluated using Spearman’s correlation coefficient and Pearson’s correlation coefficient. Statistical analyses were performed using SPSS (version 25 for Windows, IBM Japan, Ltd., Tokyo, Japan). Statistical significance was set at a *p*-value of <0.05.

## Figures and Tables

**Figure 1 ijms-26-03026-f001:**
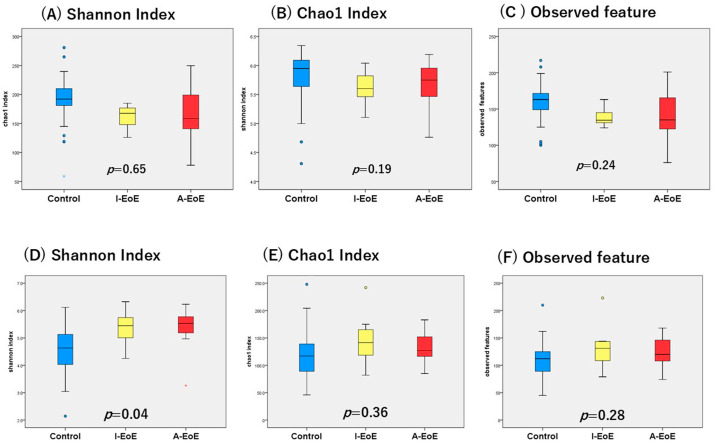
Alpha diversity indices: (**A**) Shannon index, (**B**) Chao1 index, and (**C**) observed feature using saliva; (**D**) Shannon index, (**E**) Chao1 index, and (**F**) observed feature using esophageal mucus.

**Figure 2 ijms-26-03026-f002:**
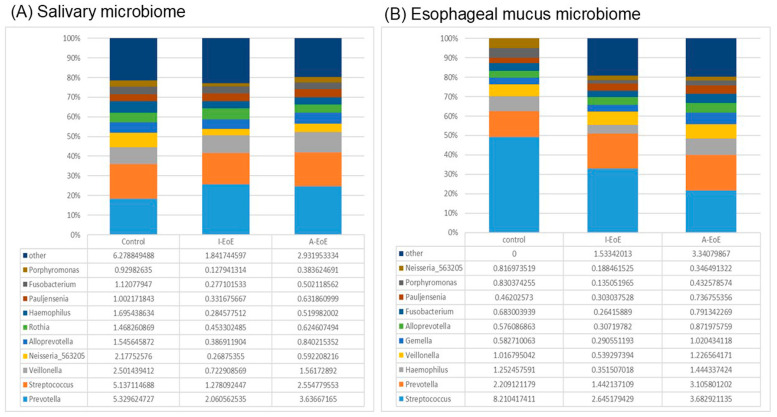
Taxonomic composition of saliva (**A**) and esophageal mucus (**B**): top 10 genera in the bacterial flora.

**Figure 3 ijms-26-03026-f003:**
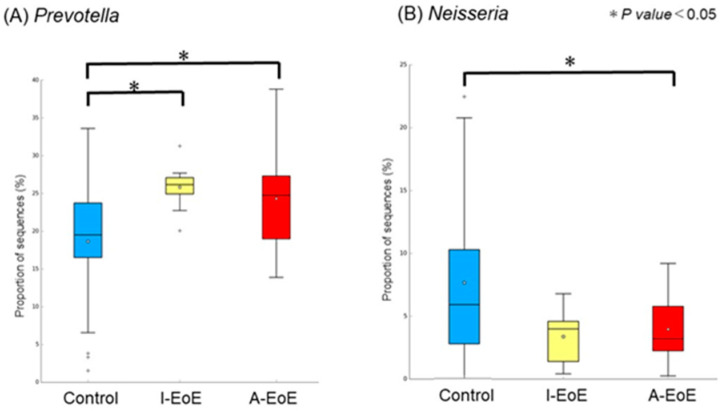
(**A**,**B**) Comparison of the esophageal mucus microbiota among the control, inactive (I)-EoE, and active (A)-EoE groups in saliva: (**A**) *Prevotella* and (**B**) *Neisseria*. (**C**–**E**) Comparison between the two groups in the salvia microbiota: (**C**) control vs. I-EoE, (**D**) control vs. A-EoE, and (**E**) I-EoE vs. A-EoE.

**Figure 4 ijms-26-03026-f004:**
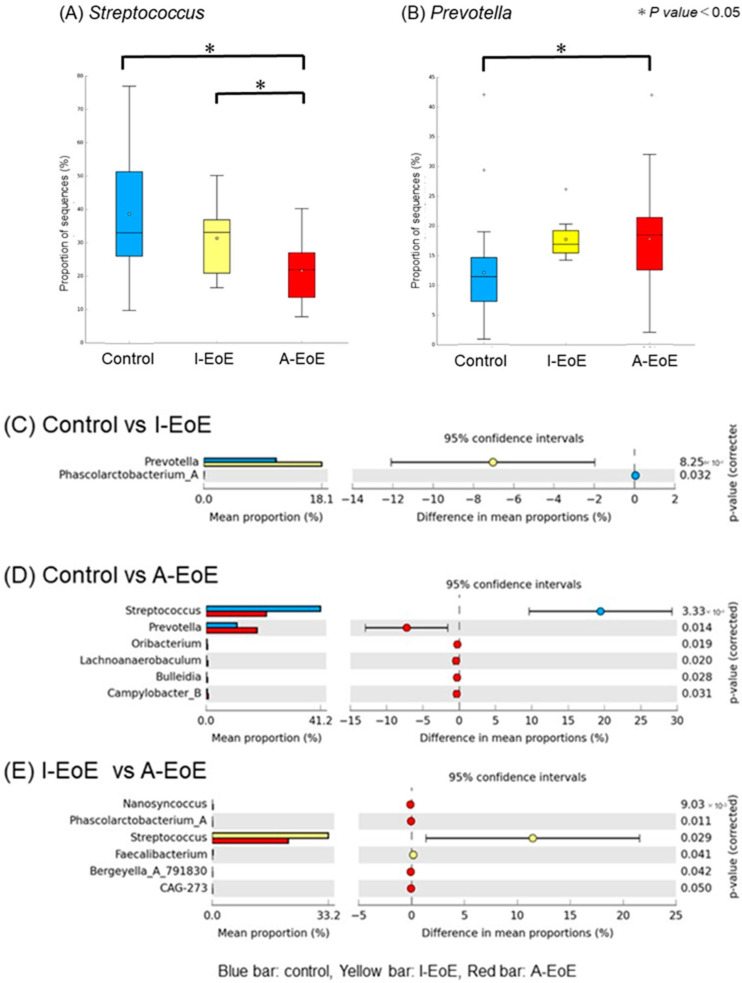
(**A**,**B**) Comparison among the control, inactive (I)-EoE, and active (A)-EoE groups of the esophageal mucus microbiota: (**A**) *Streptococcus* and (**B**) *Prevotella* spp. (**C**–**E**) Comparison of the esophageal microbiota between the two groups: (**C**) control vs. I-EoE, (**D**) control vs. A-EoE, and (**E**) I-EoE vs. A-EoE.

**Figure 5 ijms-26-03026-f005:**
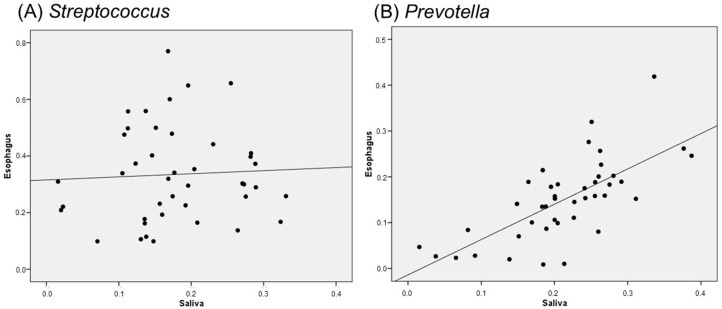
Correlation between salivary and esophageal microbiota: (**A**) *Streptococcus* and (**B**) *Prevotella*.

**Figure 6 ijms-26-03026-f006:**
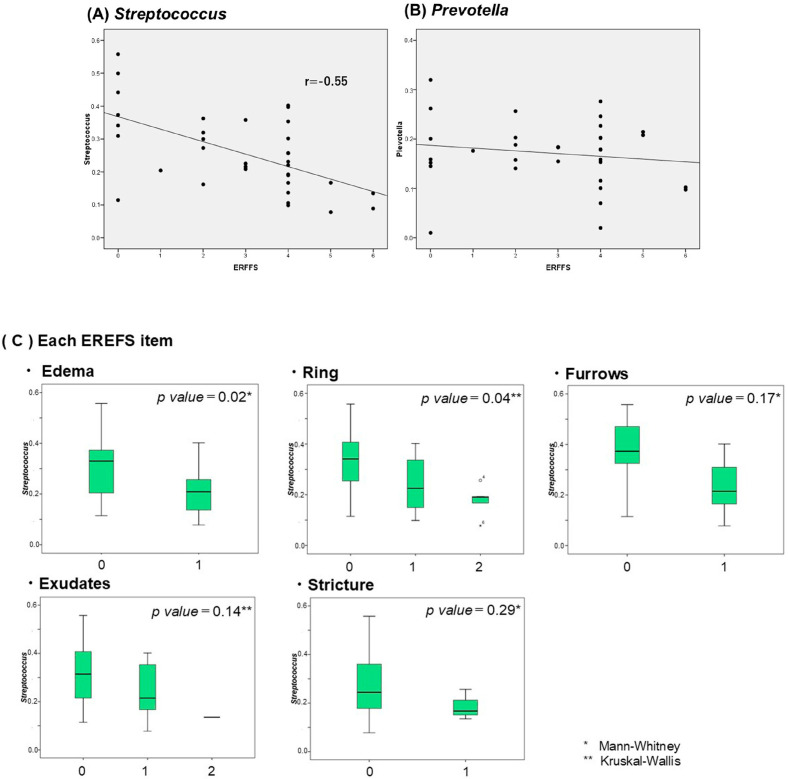
Correlation of esophageal bacteria flora and eosinophilic esophagitis endoscopic reference score (EREFS): (**A**) *Streptococcus* and (**B**) *Prevotella*. (**C**) Correlation of streptococcus and each item of EREFS.

**Figure 7 ijms-26-03026-f007:**
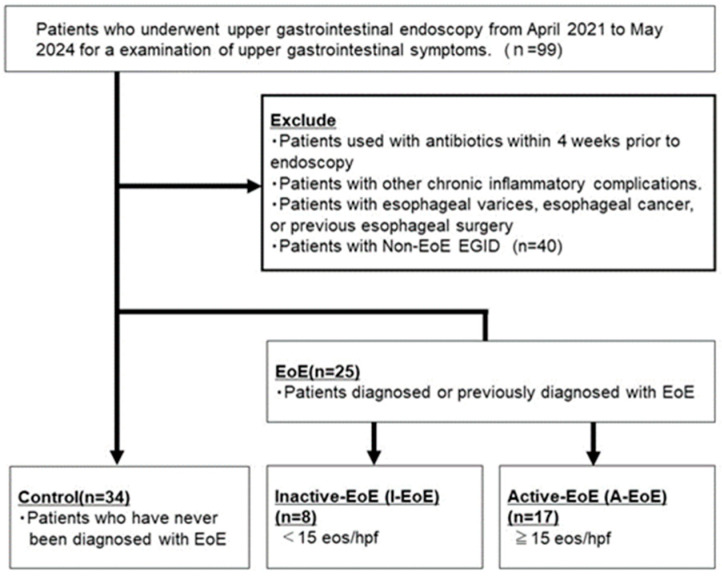
**Study design:** Three groups: control, inactive EoE (I-EoE), and active EoE(A-EoE).

**Table 1 ijms-26-03026-t001:** Patients characteristics.

	Control (n = 28)	EoE		*p-Value* *3 Groups*	*p-Value*
		Inactive (n = 8)	Active (n = 17)		*2 Groups* *Cont vs. EoE*
Gender (M)	11 (39%)	5 (62%)	10 (59%)	0.3	0.52
Age (mean ± SD; yr)	49 [29–62]	52 [37–66]	50 [42–57]	0.9	0.23
Allergic comorbidities (n, %)					
Asthma	0	3 (38%)	5 (29%)	0.04	<0.01
Eczema	1 (0%)	6 (75%)	3 (18%)	<0.01	<0.01
Allergic rhinitis	6 (21%)	2 (25%)	12 (71%)	0.01	0.05
Food allergies	6 (21%)	4 (40%)	11 (65%)	0.02	<0.01
Clinical symptoms (n, %)					
Dysphagia	4 (14%)	6 (75%)	13 (76%)	<0.01	<0.01
Abdominal pain	9 (32%)	2 (25%)	3 (18%)	0.32	0.2
Nausea, vomiting	1 (0%)	0	5 (29%)	0.05	0.08
Endoscopy findings (n, %)					
Edema	-	0 [0–0]	1 [1–1]	<0.01	
Rings	-	0 [0–1]	1 [1–2]	<0.01	
Furrows	-	0 [0–1]	1 [1–1]	<0.01	
Exudates	-	0 [0–1]	1 [1–1]	<0.01	
Structure	-	0 [0–0]	0 [0–0]	0.28	
EREFS	-	2 [2–0]	4 [4–4]	0.02	
The number of eosinophilic infiltration (median IQR: eos/hpf)	-	0 [0–20]	65 [24–85]	<0.01	
Medication (n, %)					
Proton pump inhibitors	6 (21%)	7 (88%)	4 (24%)	0.59	0.27
Topical steroids	0 (0%)	2 (25%)	2 (12%)	<0.01	0.04
Antiallergic medicine	0 (0%)	6 (75%)	7 (41%)	<0.01	<0.01
*H. pylori* infection (uninfected/ exposed/currently Infected)	22/5/1	5/3/0	14/3/0	0.85	0.6
*Cytokines expression in esophageal mucosa*					
*IL-4*	12.5 (2.3–111.3)	240.2 (159.4–333.7)	8.4 (5.6–12.4)	0.03	0.53
*IL-13*	1.0 (0.53–17.1)	19.6 (7.9–29.4)	1.5 (0.74–2.4)	0.02	0.26
*IL-33*	1.7 (1.0–12.6)	26.4 (4.8–84.9)	3.2 (2.9–3.3)	0.02	0.25
*TGF-beta1*	2.0 (1.1–6.1)	3.5 (2.0–5.2)	4.0 (2.4–9.7)	0.65	0.19
*TSLP*	2.9 (1.8–13.5)	8.9 (5.5–15.5)	3.4 (2.3–4.6)	0.58	0.80
*CCL26*	1.4 (0.97–6.7)	42.0 (21.2–113.4)	0.38 (0.11–0.49)	0.01	0.11
*CXCL16*	1.0 (0.72–1.9)	6.0 (2.5–25.2)	1.2 (0.69–2.4)	0.03	0.76

Three groups: control, I-EoE, and A-EoE; two groups: control vs. EoE (I-EoE + A-EoE). EREFS: eosinophilic esophagitis endoscopic reference score

**Table 2 ijms-26-03026-t002:** PCR primer sequences.

Target Gene		Sequence
IL-4	FP	ACATCTTTGCTGCCTCCAA
	RP	AGGCAGCGAGTGTCCTTCT
IL-13	FP	GATTCTGCCCGCACAAGGT
	RP	CGATTTTGGTGTCTCGGACAT
IL-33	FP	GGCTGAGAATTACCATACAAGG
	RP	AGTGTTTTTCAGATGGGATGA
TGFβ1	FP	CGCGTGCTAATGGTGGAAA
	RP	TGTGTGTACTCTGCTTGAACTTG
CCL26	FP	TCCCTCCTGAGTCTCCACCTT
	RP	CTGTATTGGAAGCAGCAGGTCTT
CXCL16	FP	CCTATGTGCTGTGCAAGAGGAG
	RP	CTGGGCAACATAGAGTCCGTCT

## Data Availability

The raw data supporting the conclusions of this article will be made available by the authors upon request. The data were restricted for privacy reasons.

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
