# Peer review of "The Relationship Between Bacterial Flora in Saliva and Esophageal Mucus and Endoscopic Severity in Patients with Eosinophilic Esophagitis"

_ijms, 2025, doi:10.3390/ijms26073026_

Round 1
Reviewer 1 Report
Comments and Suggestions for Authors
This study investigates the bacterial composition of saliva and esophageal mucus in patients with eosinophilic esophagitis (EoE) and examines potential correlations with disease activity and mucosal cytokine expression. The authors utilize 16S rRNA sequencing to characterize microbial diversity and composition, comparing findings across active EoE, inactive EoE, and non-EoE controls.
While the study addresses an interesting and relevant topic, there are several concerns related to methodology, statistical robustness, and clinical relevance, which must be addressed before publication. The authors should cite DOI: 10.3390/jcm13133805, as it provides important insights into the microbiota's role in eosinophilic esophagitis and should be included to strengthen the discussion section.
Major Comments
- The authors should cite DOI: 10.3390/jcm13133805, as it provides important insights into the microbiota's role in eosinophilic esophagitis and should be included to strengthen the discussion section.
- The study includes only 59 patients, subdivided into 8 inactive EoE, 17 active EoE, and 28 controls. This small sample size limits the generalizability of the results and increases the risk of type II errors. The study is single-center, which reduces external validity. The authors should acknowledge these limitations more explicitly in the discussion and propose future multicenter validation studies.
- The study characterizes bacterial composition but does not assess functional microbiota changes, such as metabolic activity or microbial interactions with the host immune system. There is no data on whether microbiota changes correlate with treatment response or disease progression over time. The authors should acknowledge this limitation and discuss potential functional studies (e.g., metagenomic or metabolomic approaches) better to understand the role of microbiota in EoE pathogenesis.
- The study reports significant differences in Prevotella, Neisseria, and Streptococcus between EoE and controls, but does not explore the biological implications of these findings in detail. The role of salivary microbiota as a potential biomarker for EoE or as a driver of disease pathogenesis is not sufficiently discussed. The discussion should explore whether salivary microbiota alterations could serve as a diagnostic or prognostic tool for EoE.
- Several factors could influence microbiome composition, including dietary habits, antibiotic use, proton pump inhibitor (PPI) use, and genetic predisposition. The study does not control for these confounders, making it difficult to determine whether the observed microbiota differences are causative or incidental. The authors should discuss how confounding factors may have influenced their results and suggest potential future approaches to control for these variables.
Minor Comments
-
- The manuscript needs grammatical improvements and better sentence structure for readability.
- Example: "The most common genera detected were Streptococcus, Prevotella, Veillonella, and Haemophilus." Suggested revision: "The most frequently detected bacterial genera included Streptococcus, Prevotella, Veillonella, and Haemophilus."
- Confidence intervals (CIs) and effect sizes are missing in some comparisons. If applicable, the authors should report CIs for their key findings to enhance transparency.
- Figure legends should be expanded to provide a clearer interpretation of the results.
- Tables should include additional statistical information, such as p-values and effect sizes. The authors should revise figure legends for clarity and ensure tables are sufficiently detailed.
Author Response
Dear Commenter 1
Thank you for your kind advice and suggestion. I followed your advice and revised my paper. I think it has become a better paper.
Comments and Suggestions for Authors
This study investigates the bacterial composition of saliva and esophageal mucus in patients with eosinophilic esophagitis (EoE) and examines potential correlations with disease activity and mucosal cytokine expression. The authors utilize 16S rRNA sequencing to characterize microbial diversity and composition, comparing findings across active EoE, inactive EoE, and non-EoE controls.
While the study addresses an interesting and relevant topic, there are several concerns related to methodology, statistical robustness, and clinical relevance, which must be addressed before publication. The authors should cite DOI: 10.3390/jcm13133805, as it provides important insights into the microbiota's role in eosinophilic esophagitis and should be included to strengthen the discussion section.
Thank you for your kind advice. We add some papers that described about eosinophilic esophagitis and saliva and esophageal mucosal flora. Unfortunately, the paper you indicated was not related to the topic.
Major Comments
- The authors should cite DOI: 10.3390/jcm13133805, as it provides important insights into the microbiota's role in eosinophilic esophagitis and should be included to strengthen the discussion section.
Thank you for your kind advice. Unfortunately, the paper you indicated was not related to the topic. However, we add some papers that described about eosinophilic esophagitis and saliva and esophageal mucosal flora.
- The study includes only 59 patients, subdivided into 8 inactive EoE, 17 active EoE, and 28 controls. This small sample size limits the generalizability of the results and increases the risk of type II errors. The study is single-center, which reduces external validity. The authors should acknowledge these limitations more explicitly in the discussion and propose future multicenter validation studies.
Thank you for your remarks. We have added a note regarding the limitations of this study that you pointed out in line 281-284. “First, the number of patients was small. The small sample size limits the generalizability of the results and increases the risk of type II errors. Second, the study is single-center, which reduces external validity. We should study future multicenter validation studies."
- The study characterizes bacterial composition but does not assess functional microbiota changes, such as metabolic activity or microbial interactions with the host immune system. There is no data on whether microbiota changes correlate with treatment response or disease progression over time. The authors should acknowledge this limitation and discuss potential functional studies (e.g., metagenomic or metabolomic approaches) better to understand the role of microbiota in EoE pathogenesis.
Thank you for your suggestion. As you pointed out, we did not perform functional or metabolic analysis in this study. We have added new paragraph to the Discussion section regarding your point in line 280-292. Besides, we add this point limitation paragraph in line 297-301.
- The study reports significant differences in Prevotella, Neisseria, and Streptococcus between EoE and controls, but does not explore the biological implications of these findings in detail. The role of salivary microbiota as a potential biomarker for EoE or as a driver of disease pathogenesis is not sufficiently discussed. The discussion should explore whether salivary microbiota alterations could serve as a diagnostic or prognostic tool for EoE.
Thank you for your suggestion. We have added one paragraph to the Discussion section regarding your point in line 268-284.
- Several factors could influence microbiome composition, including dietary habits, antibiotic use, proton pump inhibitor (PPI) use, and genetic predisposition. The study does not control for these confounders, making it difficult to determine whether the observed microbiota differences are causative or incidental. The authors should discuss how confounding factors may have influenced their results and suggest potential future approaches to control for these variables.
Thank you very much for your advice. We add the new sentence about confounding factors in limitation paragraph in discussion section in line 323-334.
Minor Comments
- The manuscript needs grammatical improvements and better sentence structure for readability.
- Example: "The most common genera detected were Streptococcus, Prevotella, Veillonella, and Haemophilus." Suggested revision: "The most frequently detected bacterial genera included Streptococcus, Prevotella, Veillonella, and Haemophilus."
Thank you for your advice. We rewrite it.
- Confidence intervals (CIs) and effect sizes are missing in some comparisons. If applicable, the authors should report CIs for their key findings to enhance transparency.
Thank you for your advice. We add the sentence about Cis and effect sizes in the results section.
- Figure legends should be expanded to provide a clearer interpretation of the results.
Thank you for your suggestion. We rewrite figure legends.
- Tables should include additional statistical information, such as p-values and effect sizes. The authors should revise figure legends for clarity and ensure tables are sufficiently detailed.
Thank you for your suggestion. We rewrite figure legends.

Reviewer 2 Report
Comments and Suggestions for Authors
This manuscript provides a valuable investigation into the microbiota profiles of saliva and esophageal mucus in patients with eosinophilic esophagitis (EoE). The study is well-structured, methodologically sound, and addresses an important gap in understanding the microbial dynamics associated with EoE severity. The findings highlight significant shifts in Prevotella and Streptococcus abundance, correlating with endoscopic activity, which contributes meaningfully to the field. With minor revisions to enhance clarity and depth, this work is suitable for publication in International Journal of Molecular Sciences.
Suggestions
- Language and Clarity
The manuscript has very high plagiarism rate of 37%. please rephrase and remove this high rate of plagiarism. In abstract remove the words like background, method, results, conclusions. For IJMS format these sub titles headings in abstract are not required. A proofread by a native English speaker is recommended to refine syntax and terminology. Correct "Precotella" to Prevotella throughout (e.g., Figures 2–4, Results/Discussion). "The brush method was selected as the specimen sampling method..." → "The brush method was chosen over biopsy due to its higher bacterial DNA recovery rate.". "ERFES" → "EREFS" (Table 1, Figure 6). Clarify labels (e.g., Figure 3C–E: Specify group comparisons in captions).
- Results Section
Figure 1: Rename "Chol index" to "Chao1 index" in panels B, E. Figure 3 and 4: Simplify overlapping subpanels (C–E) or consolidate into supplementary data. Table 1: Standardize formatting (e.g., "p-value" capitalization, consistent decimal places for percentages). Ensure all figures are cited in the text (e.g., Figure 6C is mentioned but not included in the provided content).
- Discussion
Expand on how Prevotella (Th17/IL-23 axis) and Streptococcus (probiotic roles) mechanistically influence EoE. Discuss potential for Streptococcus-based probiotics or Prevotella-targeted therapies. Acknowledge that sampling only the distal esophagus may miss regional microbial variations.
- References
Verify citation years (e.g., "2025 consensus guidelines" likely intended as 2023). Ensure journal abbreviations align with ISO standards (e.g., "N. Engl. J. Med." instead of full names).
- Supplementary Materials
Include sample metadata (e.g., diet, PPI usage) to contextualize microbiota changes. Deposit sequencing data in public repositories (e.g., NCBI SRA) for reproducibility.
This study advances our understanding of microbiota dysbiosis in EoE and its correlation with disease severity. The findings are novel, and the methodology is robust. With minor revisions to address language clarity, figure annotations, and mechanistic discussion, this manuscript will be a significant contribution to the field. Therefore, I accept this after minor revision.
Comments on the Quality of English LanguageA proofread by a native English speaker is recommended to refine syntax and terminology. The manuscript has very high plagiarism rate of 37%. please rephrase and remove this high rate of plagiarism.
Author Response
Open Review
This manuscript provides a valuable investigation into the microbiota profiles of saliva and esophageal mucus in patients with eosinophilic esophagitis (EoE). The study is well-structured, methodologically sound, and addresses an important gap in understanding the microbial dynamics associated with EoE severity. The findings highlight significant shifts in Prevotella and Streptococcus abundance, correlating with endoscopic activity, which contributes meaningfully to the field. With minor revisions to enhance clarity and depth, this work is suitable for publication in International Journal of Molecular Sciences.
Thank you for your kind advice and suggestion. I followed your advice and revised my paper. I think it has become a better paper.
Suggestions
- Language and Clarity
The manuscript has very high plagiarism rate of 37%. please rephrase and remove this high rate of plagiarism.
Thank you for your kind advice. I check it and rewrite.
In abstract remove the words like background, method, results, conclusions. For IJMS format these sub titles headings in abstract are not required.
Thank you for your kind advice. We omit subtitles in abstract section.
A proofread by a native English speaker is recommended to refine syntax and terminology.
Thank you for your advice. We check English again and rewrite.
Correct "Precotella" to Prevotella throughout (e.g., Figures 2–4, Results/Discussion).
Thank you for your suggestion. We correct the words.
"The brush method was selected as the specimen sampling method..." → "The brush method was chosen over biopsy due to its higher bacterial DNA recovery rate.".
Thank you for your kind suggestion. We rewrite this sentence.
"ERFES" → "EREFS" (Table 1, Figure 6). Clarify labels (e.g., Figure 3C–E: Specify group comparisons in captions).
Thank you for your kind suggestion. We rewrite the word, and clarify labels.
- Results Section
Figure 1: Rename "Chol index" to "Chao1 index" in panels B, E.
Thank you for your suggestion. We correct the words.
Figure 3 and 4: Simplify overlapping subpanels (C–E) or consolidate into supplementary data.
Thank you for your suggestion. We correct the words.
Table 1: Standardize formatting (e.g., "p-value" capitalization, consistent decimal places for percentages).
Thank you for your suggestion. We correct the words.
Ensure all figures are cited in the text (e.g., Figure 6C is mentioned but not included in the provided content).
- Discussion
Expand on how Prevotella (Th17/IL-23 axis) and Streptococcus (probiotic roles) mechanistically influence EoE.
Thank you for your kind advice. We add new sentence about your advice, Prevotella (Th17/IL-23 axis) and Streptococcus (probiotic roles), in line .
Discuss potential for Streptococcus-based probiotics or Prevotella-targeted therapies.
Thank you for your suggestion. We add new sentence about potential for Streptococcus-based probiotics or Prevotella-targeted therapies in line - .
Acknowledge that sampling only the distal esophagus may miss regional microbial variations.
Thank you for your advice. We add the new sentence about sampling in paragraph of limitation in line - .
- References
Verify citation years (e.g., "2025 consensus guidelines" likely intended as 2023). Ensure journal abbreviations align with ISO standards (e.g., "N. Engl. J. Med." instead of full names).
Thank you very much for your suggestion. We confirm them.
- Supplementary Materials
Include sample metadata (e.g., diet, PPI usage) to contextualize microbiota changes. Deposit sequencing data in public repositories (e.g., NCBI SRA) for reproducibility.
Thank you very much for your advice. We add supplemental data about PPI usage.

Round 2
Reviewer 1 Report
Comments and Suggestions for Authors
The authors gently answered all questions and comments.
Unfortunately, I made a mistake with the DOI of the paper, and the correct one is the following:
- DOI: 10.3390/diagnostics14080858
You should cite this paper in your study for the reasons previously explicated.
Author Response
Reviewer1 Round 2
Comments and Suggestions for Authors
The authors gently answered all questions and comments.
Unfortunately, I made a mistake with the DOI of the paper, and the correct one is the following:
- DOI: 10.3390/diagnostics14080858
You should cite this paper in your study for the reasons previously explicated.
Thank you very much for your advice. We add the paper to new version.